# Calcium Signaling in Plant-Insect Interactions

**DOI:** 10.3390/plants11202689

**Published:** 2022-10-12

**Authors:** Ambra S. Parmagnani, Massimo E. Maffei

**Affiliations:** Department of Life Sciences and Systems Biology, University of Turin, Via Quarello 15/a, 10135 Turin, Italy

**Keywords:** signal transduction, herbivory, reactive oxygen species, membrane potential, oral secretions, calcium imaging, MAPK, calmodulin, CPK

## Abstract

In plant–insect interactions, calcium (Ca^2+^) variations are among the earliest events associated with the plant perception of biotic stress. Upon herbivory, Ca^2+^ waves travel long distances to transmit and convert the local signal to a systemic defense program. Reactive oxygen species (ROS), Ca^2+^ and electrical signaling are interlinked to form a network supporting rapid signal transmission, whereas the Ca^2+^ message is decoded and relayed by Ca^2+^-binding proteins (including calmodulin, Ca^2+^-dependent protein kinases, annexins and calcineurin B-like proteins). Monitoring the generation of Ca^2+^ signals at the whole plant or cell level and their long-distance propagation during biotic interactions requires innovative imaging techniques based on sensitive sensors and using genetically encoded indicators. This review summarizes the recent advances in Ca^2+^ signaling upon herbivory and reviews the most recent Ca^2+^ imaging techniques and methods.

## 1. Introduction

The plant–insect interaction dates back to prehistorical times [1]. The evolution of plant defense responses against herbivores requires stress response-associated mechanisms, including phytohormones production [2,3,4], host specificity [5], protease inhibitor activities [6,7], plant volatile emission [8], phenotypic plasticity [9], cross-species communication [10], morpho-physiological and chemical reprogramming [11], response to insect-derived elicitors or effectors [12,13], all involving metabolic costs [14].

The recognition and appropriate response to the attacking enemy occur within a few minutes and require the activation of specific pathways most of which finally lead to gene activation (the signaling pathway) [15]. Upon insect feeding on plants, rapid early events eventually lead to the production and release of plant defense molecules [16]. These signaling events also function in propagation of long-distance signals (calcium, reactive oxygen species, and electrical signals), which contribute to rapid, systemic induction of defense responses [17]. Interestingly, the speed of response to biotic damage is proportional to the speed of feeding. For instance, chewing herbivores induce faster responses that sucking herbivores, which in turn are faster in inducing plants responses when compared to microorganisms [18]. However, the term rapid is relative to the communication systems of plants which by no way is comparable or similar the to highly efficient and rapid nervous system of animals, as recently clarified [19]. Therefore, biotic stresses induced by herbivores result in diverse physiological changes in plants. 

### Generation of Calcium Signals

One of the earliest players in plant–insect interactions is calcium (Ca^2+^), a universal second messenger the intracellular variation in concentration of which has been associated to the plant perception of biotic stress [20,21]. Herbivory-induced cytosolic Ca^2+^ concentration ([Ca^2+^]_cyt_) elevation involves multiple channels and pathways regulating local and long-distance [Ca^2+^]_cyt_ signals [22]. The depolarization of the plasma membrane potential (Vm) is another early event induced by insect feeding on leaves, and results from both the damage caused by the insect and the delivery of oral secretions (OS) [23]. Although this herbivore-induced Vm depolarization depends on a Ca^2+^-dependent opening of potassium ion (K^+^) channels [24], the attacked leaf remains depolarized for an extended period, which cannot be explained by the sole action of K^+^ channels. The plant plasma membrane H^+^-ATPase is strongly inhibited by insect OS, and this inhibition contributes to the log-lasting Vm depolarization [25]. Studies on mechanical and robotic wounding of leaves show that these treatments do not induce a Ca^2+^ signaling, whereas significant increase in [Ca^2+^]_cyt_ were observed only after herbivory [26].

Herbivore attacks can trigger Ca^2+^ waves traveling a long distance to transmit and convert the local signal to a systemic defense program in the whole plant, and recent studies uncovered Ca^2+^ channels that orchestrate specific Ca^2+^ signatures [27]. Electrical signals can result in systemic Ca^2+^ and reactive oxygen species (ROS) waves, as demonstrated in several insect herbivores such as *Spodoptera littoralis*, *Pieris brassicae*, *Myzus persicae* and *Tetranychus urticae* feeding of model plants like *Arabidopsis thaliana*; important crops like *Hordeum vulgare*, *Phaseolus lunatus*, *Nicotiana tabacum* and *Vicia faba*; the living fossil plant *Ginkgo biloba* and the fern *Pteris vittata* [18,24,28,29,30,31,32].

Mutations affecting Ca^2+^ homeostasis interfere with the plant’s leaf-to-leaf electrical signaling capacity as shown when Arabidopsis type IIB auto-inhibited Ca^2+^-ATPases (ACAs) was used as a candidate gene for a reverse genetic screening [33]. Both root-to-shoot Ca^2+^ waves and slow wave potentials (SWPs) are triggered by either root wounding or the application of glutamate (Glu) to wounded roots. Ca^2+^ waves and SWPs are dependent on the activity of glutamate receptor-like proteins (GLRs) and P-type H^+^-ATPase AHA1. Inhibition of the H^+^-ATPase activity due to herbivore OS [25] induces the alkalinization of the apoplast that depolarizes Vm. The activation of GLRs, along with the release of Glu from damaged phloem, causes the Vm depolarization in the form of SWPs that together with the [Ca^2+^]_cyt_ increase propagates the systemic wound signaling [34,35]. Recently, a key functional role for the cyclic nucleotide-gated ion channel CNGC19 in Arabidopsis defense against *Spodoptera litura* demonstrated the role of Ca^2+^-mediated defense signaling, with the modulation of late events like the biosynthesis of phytohormones and secondary metabolites [36].

Wounding may also induces a drop in hydraulic pressure (turgor pressure, Ψ_T_) and an increase in apoplastic amino acids (AAapo), including Glu, that are perceived by plasma-membrane located mechanosensitive ion channels (MSC) that depolarize the plant Vm [37]. Finally, herbivore’s OS may contain oligosaccharides that interact with specific plant plasma membrane receptors [24]. For instance, spider mites secrete elicitors (tetranin1 and tetranin2) that trigger [Ca^2+^]_cyt_, ROS production and Vm depolarization [12]. In Arabidopsis, plasma membrane localized NADPH oxidases respiratory burst oxidase homolog D (RBOHD) functions as an essential regulator of ROS [38]. Herbivore-produced elicitors like *N*-linolenoyl-l-glutamine (GLN18:3) induce a RBOHD and RBOHF-dependent ROS burst in several plant species [13,39].

Figure 1 summarizes the early events upon herbivory where Ca^2+^ plays a central role along with Vm depolarization and ROS production.

## 2. Calcium and ROS Signaling

The plant shows a consistent number and variability of signaling molecules throughout, but three potentially interacting messengers: ROS, Ca^2+^ and electrical signaling are interlinked to form a network supporting rapid signal transmission [40,41,42,43].

Several plant species produce hydrogen peroxide (H_2_O_2_) in response to herbivory. The expression of the Ca^2+^-sensing aequorin system in transgenic soybean (*Glycine max*) suspension cells and the effects of herbivore-wounding in Lima bean (*P. lunatus*) leaves show increasing amounts of H_2_O_2_ that correlate with a higher [Ca^2+^]_cyt_ [44]. The same responses were found in the living fossil *G. biloba* leaves damaged by *S. littoralis* [30] and in the fern *P. vittata* [29]. Changes of [Ca^2+^] fluxes and H_2_O_2_ were detected in a tobacco variety (G140) infested by the tobacco aphid *M. persicae*. Even in this case, H_2_O_2_ accumulation depended on Ca^2+^ influx, which were both increased in a long period of aphid feeding [45].

Ca^2+^ and ROS signaling are induced by herbivore-associated molecular patterns (HAMPs) [46]. As shown in Figure 1, ROS signals could activate the Ca^2+^ permeable channels in membranes [47]; ROS and Ca^2+^ signaling are the consequence of the combined effect of wounding and the secretion of salivary elicitors from the feeding insect [48,49,50]. For instance, aphids like *Acyrthosiphon pisum* produce two types of saliva that mediate their interactions with plants which are based on Ca^2+^ signaling-related responses [51]. The activity of *S. littoralis* ventral eversible gland contains elicitors able to trigger both Ca^2+^ signaling and H_2_O_2_ production in Arabidopsis [52]. The piercing-sucking planthopper (*Laodelphax striatellus*) secretes salivary proteins into its plant host during feeding that enhance plant resistance to insects by inducing Ca^2+^ and ROS signaling pathways [53]; whereas the tobacco hornworm caterpillar (*Manduca sexta*) OS induce Ca^2+^-dependent ROS production in tomato (*Solanum lycopersicum*) protoplasts [54]. Elicitors produced during the feeding activity of the plant-sucking arthropod herbivore two-spotted spider mite (*T. urticae*) were found to trigger the [Ca^2+^]_cyt_ influx and the generation of ROS in kidney bean plants (*P. vulgaris*) [12].

Besides OS direct activity, several HAMPs released upon cell damage trigger cascades of events, eventually leading to a coordinated response. Among these, extracellular DNA (eDNA), deriving from cell disruption made by herbivory, induces an increase in [Ca^2+^]_cyt_; this event is associated to the opening of K^+^ channels, with particular action on ligand-gated rectified K^+^ channels, which in turn are correlated to ROS production [55,56]. It has been suggested that fragmented DNA might play a crucial role as an important and powerful elicitor involved in early and late responses to biotic stress [56]. Extracellular ATP (eATP) is a constitutive HAMP that is released by wounding and herbivory. eATP can increase [Ca^2+^]_cyt_, and Arabidopsis loss of function mutants were found to have an impaired increase in [Ca^2+^]_cyt_ in response to eATP, thus demonstrating a link between eATP and Ca^2+^ signaling [57].

Calcium and ROS signaling are also induced by volatile organic compounds (VOCs) [15]. Plants respond to herbivory by emitting VOCs composed of both green leaf volatiles (GLVs) and terpenoids, which are released into the surrounding atmosphere and are perceived by receiving plants [58]. GLVs emitted by herbivore-wounded tomato (including (*Z*)-3-hexenyl acetate) were found to induce a strong [Ca^2+^]_cyt_ increase in receiving plants [59]. The emission of β-ocimene induced by (*Z*)-3-hexenol, linalool, α-farnesene and (*E*)-4,8–dimethyl–1,3,7-nonatriene (DMNT) (which are typical VOCs emitted upon herbivory) were dependent on Ca^2+^ signaling [60]. H_2_O_2_ and [Ca^2+^]_cyt_ increased in Arabidopsis mesophyll cells after being treated with the volatile monoterpene alcohol linalool in the attempt to increase plant resistance to diamondback moth (*Plutella xylostella*) [61].

Other interesting effects of plant–insect interactions on Ca^2+^ and ROS signaling were found when (*Z*)-11-hexadecenal, the main component of *P. xylostella* female sex pheromone, was tested in *Brassica nigra*. The sex pheromone induced depolarization of the Vm, an increase in [Ca^2+^]_cyt_ and production of H_2_O_2_, demonstrating the ability of plants to detect and respond to volatiles emitted by non-plant organisms [62].

## 3. Calcium Binding Proteins (CBPs)

To activate the appropriate cell response and in order to become informative, the Ca^2+^ message needs to be decoded and relayed. This task is brought by Ca^2+^-binding proteins (CBPs) [63]. Most CBPs are characterized by the presence in their sequence of the canonical Ca^2+^-binding motif called the EF-hand [64] (see Figure 2). CBPs in the saliva of herbivorous insects function as effectors to attenuate host plant defenses and thus improve insect feeding performance [65]. A putative EF-hand Ca^2+^-binding protein (LsECP1) in the small brown planthopper *Laodelphax striatellus* exhibits Ca^2+^-binding activity. Knocking down LsECP1 in *L. striatellus* prompted a higher level of [Ca^2+^]_cyt_ in rice fed plants, whereas overexpression suppressed wound-induced H_2_O_2_ accumulation [66]. 

CBPs are also involved in the transmission of devastating viruses into plant phloem. Viral infection may inhibit CBP expression and this event facilitates filament-mediated viral secretion first into salivary cavities and then into plant phloem which in rice causes an increased [Ca^2+^]_cyt_, followed by a substantial H_2_O_2_ production [67]. 

The NlSEF1 protein is a CBP highly expressed in the salivary glands of the brown planthopper *Nilaparvata lugens*. It has an EF-hand Ca^2+^-binding activity and can be secreted into rice plants when infested by the planthopper. Knocking down NISEF1 in the planthopper elicited higher levels of [Ca^2+^]_cyt_ and H_2_O_2_ in rice, indicating that NlSEF1 functions as an effector and plays important roles in interactions between the planthopper and rice by mediating the plant’s defense responses [68,69].

### 3.1. Calmodulin (CaM) and Calmodulin-like (CML)

Calmodulin (CaM) is present in all eukaryotic cells and is the prototype of a CBP [70]. Being involved in the Ca^2+^ signaling pathway, CaM acts by modifying its interactions with various CaM-binding proteins [20]. A divergent form of CaMs (CaM-like proteins, CMLs) is also present in plants [71,72].

The salivary glands of planthoppers *N. lugens* and *Laodelphax striatellus* show high expression of CaMs that are secreted into the rice plants during feeding and CaM-silenced planthopper nymphs elicited relatively high levels of H_2_O_2_ and callose accumulation in rice plants [73]. The transcriptome of spinach leaves exposed to beet armyworm (*Spodoptera exigua*) larvae identified four CAMs (CAM3, CAM5-1, CAM6-1, and CAM6-2) which were associated with Ca^2+^ signaling. The genes coding for these CAMs were highly co-overexpressed with several endoplasmic reticulum-type Ca^2+^-transporting-ATPase genes, known to be involved in Ca^2+^ transport, confirming also the important role of the Ca^2+^ signaling cascade in the defense against insect attack [74]. In Arabidopsis, calmodulin 1 (CAM1) *cam1* mutants were more resistant to *S. littoralis* than in the wild-type Arabidopsis group [75]; whereas, calmodulin 3 (CAM3) participated in Ca^2+^-ATPase activation [61]. CAM3 was also found to interact with the *N*-terminus of Ca^2+^-ATPase isoform 8 (ACA8), and treatment with the monoterpene linalool of Arabidopsis leaves showed the enhancement of some JA-related genes and defense genes expressions [61]. *cam1* and *wrky53* mutants are more resistant to *S. littoralis* than the wild-type Arabidopsis. The high [Ca^2+^]_cyt_ causes the breaking down of the CAM1-WRKY53 complex, then the detachment of WRKY53 reduces the JA content by downregulating the LOXs gene expression [75]. CaM can both positively and negatively regulate cyclic nucleotide-gated channels (CNGCs) [76], which are involved in jasmonate-induced Ca^2+^ mobilization [77], a typical response to herbivore attack [78]. 

Plants contain unique Ca^2+^-sensing proteins, the calmodulin-like-proteins (CML), which are involved in many stresses and developmental responses [72]. In soybean, feeding by *S. litura*, treatment with signaling compounds and wounding induce the differential expression patterns of CMLs, indicating their involvement in Ca^2+^ signaling and plant defense during herbivory [79]. In defense against herbivory, Arabidopsis CML37 positively regulates the plants’ defense against *S. littoralis* [80], whereas CML42 shows a downregulating activity and might play a role as a Ca^2+^ sensor by displaying diverse functions in responses to both abiotic stress and insect herbivory [81]. Thus, during herbivory, plants may use CML37 and CML42 as Ca^2+^ sensor proteins to maintain equilibrium and adjust the signaling and downstream responses [82]. *S. littoralis* feeding induces Arabidopsis CML9 upon wounding and feeding; however, CML9 loss-of-function mutant lines were not affected by herbivory and the same occurred in overexpressing Arabidopsis lines. This indicates that in Arabidopsis CML9 acts more like a specialized rather than a general regulator of stress responses [78].

### 3.2. Calcium-Dependent Protein Kinases (CPKs)

In contrast to CaMs and CMLs, Ca^2+^-dependent protein kinases (CPKs or CDPKs) are distinctive Ca^2+^ decoders because of their characteristic of having both downstream signal propagation capabilities and a Ca^2+^ sensing component. They comprise a variable *N*-terminal part, a kinase domain and an activation domain [83]. The Ca^2+^ binding of the CaM-like domain prompts a conformational change that dislocates the pseudo-substrate from the kinase; this allows the occurrence of downstream phosphorylation events [84]. Ca^2+^-binding sensory proteins such as CPKs are involved in responses to herbivory following [Ca^2+^]_cyt_ variations [15,85,86], and the involvement of CPKs in response to herbivore feeding has been reviewed [87]. Transgenic *N. attenuata* plants, in which two CPKs were silenced, when attacked by *M. sexta* larvae induced high levels of defense metabolites that slowed the insect growth, demonstrated the critical roles of CPKs in modulating phytohormone (jasmonic acid) homeostasis and highlighted the complex coupling between phytohormones (e.g., jasmonate) and MAPK signaling [88]. In apple (*Malus* × *domestica*), 30 CPK genes (MdCPK) were identified and correlated to the Ca^2+^ signaling of resistant cultivars as compared to susceptible cultivars [89]. In soybean, CPKs’ transcript levels changed after wounding and exhibited specific expression patterns upon simulated *S. exigua* feeding or soybean aphid (*Aphis glycines*) herbivory, by revealing an interesting role of CPKs in soybean-insect interactions [90].

### 3.3. Calcineurin B-like (CBL)

Calcineurin B-like proteins (CBLs) are a family of Ca^2+^ sensor proteins present in plants that bind Ca^2+^ through EF-hand motifs (see Figure 2) [91,92,93]. The CBL family is unique as well. CBL-interacting protein kinase (CIPK) are downstream kinases that activate specific targets and transduce signals [94]. Changes in Ca^2+^ levels are detected by CBLs that, upon Ca^2+^ binding, modify their conformation with the subsequent interaction and activation of downstream CIPKs and CBL [95]. CBL-CIPKs play a major role in plant responses to abiotic stress [96] (e.g., salt stress) and are involved in phytohormone-, ion homeostasis- and sucrose homeostasis-related crosstalks between plant development and stress adaptation [97]. CBLs are involved also in biotic stress. Cells of *N. benthamiana* leaves overexpressing sugarcane CBL genes after inoculation with the tobacco pathogen *Ralstonia solanacearum* show that overexpression of CBL genes can effectively promote resistance to infections in tobacco plants [98]. After exposure to the pathogenic fungus *Sclerotinia sclerotiorum*, CBL1 overexpressing plants display better performance under unfavorable stress conditions and ectopic expression of CBL1 enhanced biotic stress tolerance by facilitating scavenging of the Na^+^ and ROS from the cell [99]. CBL-CIPKs have been demonstrated to play a role also in plant responses to herbivory [100].

### 3.4. Annexins (ANNs)

Annexins are proteins widely present in living organisms that bind to membrane phospholipids in a Ca^2+^-dependent manner and are involved in the regulation of plant growth and response to environmental stimuli [47]. Evidence has been provided for a contribution of ANNs in Ca^2+^ transport [22] and increasing [Ca^2+^]_cyt_ enhances the binding of annexin to plasma membrane [101], thus identifying ANNs as Ca^2+^ sensors or effectors in [Ca^2+^]_cyt_-dependent processes [102]. In pepper (*Capsicum annuum*) and tomato (*Solanum lycopersicum*), the expression of ANNs was found to be induced by biotic stress, including thrip infestation [103]. Resistant pepper plants attacked by the insect herbivore *Bemisia tabaci* were characterized by an increased expression of annexin D4-like (ANN4) [104]. In Arabidopsis, overexpression of ANN1 and ANN4 were found to increase the plant resistance to root knot nematodes [105], while ANN1 overexpressing plants induced *S. littoralis* larvae to gain less weight, confirming the importance of ANN1 in plant–insect interactions with a specific role in systemic rather than local defense responses [22]. Besides their role as CBP, ANNs may be able to act as peroxidase [106] with a phosphorylation/dephosphorylation regulation [107], in agreement with the suggestion that ANNs might provide a molecular link between ROS and [Ca^2+^]_cyt_ in the systemic defense-related signaling in plants [22]. Indeed, *AtAnn1* mutants have reduced Ca^2+^ signature upon H_2_O_2_ treatment, suggesting an important role of ANNs in ROS-induced Ca^2+^ signatures [108].

Figure 2 shows the Ca^2+^ binding to EF-hand motif and summarizes the CBPs involved in Ca^2+^ signaling.

## 4. Mitogen-Activated Protein Kinases (MAPKs)

Elicitor perception that induces Ca^2+^ influx, ROS and nitrogen reactive species (NRS) is followed by the activation of other major players in plant–insect interaction: mitogen-activated protein kinases (MAPKs) [109,110]. In *Nicotiana attenuata*, MAPKs play central roles in modulating herbivory-induced phytohormone and anti-herbivore secondary metabolites, thus regulating wounding- and herbivory-induced responses [111,112,113]. Several MAPK genes have been identified from chickpea (*Cicer arietinum*) in response to infestation by *Helicoverpa armigera* [114,115]; whereas two MAPKs were identified in soybean (*Glycine max*) attacked by the green stink bug (*Nezara viridula*) [116]. The expression of a MAPK in rice (*OsMKK3*) was induced by both mechanical wounding and the infestation of brown planthopper *N. lugens* [117], whereas RNA-Seq analyses of Balsas teosinte (*Zea mays* ssp. *parviglumis*) attacked by the armyworm (*Mythimna separata*) revealed up-regulation of genes markedly enriched in MAPK cascade-mediated signaling pathway [118]. In the date palm (*Phoenix dactylifera*) infested by the Dubas bug (*Ommatissus lybicus*), transcriptome analysis revealed the differentially expressed genes mostly belonged to calcium and MAPKs signaling pathways [119]. However, activation of MAPKs has been demonstrated also with the sole mechanical wounding [28,120].

## 5. Calcium Imaging

A breakthrough in the field of Ca^2+^ imaging was the development of sensitive fluorescent Ca^2+^ indicators, dyes and buffers by Tsien in 1980 [121]. Reporter dyes (small organic molecule-based sensors) are useful tools to measure Ca^2+^ levels without the need to genetically transform plants. This allows Ca^2+^ imaging also in those plants for which it is difficult obtaining transgenic lines [122]. In the last paragraph of this review, we focus on the development of new sensors used for Ca^2+^ imaging in plant–insect interactions.

The use of synthetic dyes (e.g., Calcium Orange) allowed the study the effect of insect chewing on leaves, allowing to observe that Ca^2+^ release was localized at the very edge of the bite (see Figure 3) [28]. Although reporter dyes have been foundational in developing the field of Ca^2+^ imaging, the advent of fluorescent protein (FP) technologies, and the associated possibility of engineering genetically encoded indicators, led to new interesting results [122,123]. In biology, a myriad of signaling processes is quantified by the use of fluorescent biosensors, which are fluorescent molecules introduced into an organism to monitor some parameters of some biological activity [124]. The development of different fluorescent sensors enabled the single cell, tissue, organ and even whole plant visualization of Ca^2+^ signals [125]. In plant studies, the bioluminescent protein aequorin (derived from the jellyfish *Aequorea victoria*) was first used in 1967 as a genetically encoded Ca^2+^ biosensor [126]. However, despite the use of aequorin to detect Ca^2+^ changes in response to various stresses in plants [127], including temperature [128], pathogens [129], salt stress [127] and wounding [130] was successful, this system it is not well-suited for real-time imaging due to the extremely low luminescent signal it produces [131]. Anyway, aequorin-expressing transgenic plants have been used in the study of diverse microbe- or damage-associated molecular patterns (MAMPs/DAMPs) where comparative analyses between the elicitors flg22 and elf18 revealed differences in Ca^2+^ signaling and cellobiose exposure which generated an intracellular Ca^2+^ elevation [129]. They have also been instrumental to quantify the response to insect OS [28]. Indeed, the study of these Ca^2+^ elevations has been largely restricted to the use of elicitors as opposed to living organisms [132].

We can devise the fluorescence-based genetically encoded indicators into two classes: (i) intensiometric single-FP biosensors and (ii) ratiometric FRET-based biosensors [125]. Although ratiometric FRET-based sensors are quantitatively accurate [133], intensiometric Ca^2+^ indicators, including GCaMP3 [134] and R-GECO1 [135], provide both higher temporal resolution and ease of use due to their generally brighter Ca^2+^-responsive signal and their simpler microscope requirements [122,125]. Comparing the ratiometric fluorescence-based Ca^2+^ sensor Yellow Cameleon NES-YC3.6 [136] and the intensity-based sensor R-GECO1, it has been demonstrated that the latter shows a significantly enhanced signal change in response to several stimuli [135]. Examples of the greater sensitivity of R-GECO1 have been provided int the study of Ca^2+^ signals in flg22- and chitin-induced experiments on a cellular scale. Here the use of GECO1 allowed to visualize well defined [Ca^2+^]_cyt_ oscillations in both epidermal and guard cells [135].

In the last decade, the approach to Ca^2+^ imaging using genetically encoded indicators had a great impact in the study of Ca^2+^ signaling, and a good amount of protocols and review papers described how to perform plant Ca^2+^ imaging in vitro but especially in vivo [20,133,134,137,138,139,140,141,142,143,144,145]. These fluorescent-based genetically encoded probes have been successfully used to study Ca^2+^ signaling in the detection of abiotic stresses [35]. In fact, the Ca^2+^ involvement in response to abiotic stimuli such as wounding is well documented [46], but its role in plant defense strategies to insect herbivory and other pathogen attacks is less understood [146]. *S. littoralis* feeding on an Arabidopsis plants expressing the GCaMP3 fluorescent protein–based [Ca^2+^]_cyt_ sensor showed a rise in [Ca^2+^]_cyt_ at the wounding position within 2 sec [134]. Furthermore, the signal spread to distal leaves in 1 to 2 min with a more pronounced signature in the vasculature, especially when the feeding activity was affecting the major vein. The propagation of this [Ca^2+^]_cyt_ signal was observed moving from younger to older leaves and vice versa [35]. By following the protocol of Toyota [35], we observed in Arabidopsis expressing the genetically encoded R-GECO1 sensor an increase in the [Ca^2+^]_cyt_ upon *S. littoralis* feeding on a rosette leaf. The signal propagates quickly through the vasculature (Figure 3 and Appendix A). Thus, it is possible to investigate the complexity of Ca^2+^ signaling with the use of reporter dyes and especially with the more recent genetically encoded Ca^2+^ indicators, which can be very helpful to decode both abiotic and biotic responses in plants.

## 6. Conclusions

Ca^2+^ variations are among the earliest events during plant–insect interactions. Ca^2+^ waves arise by the wounded plant to transmit at long distance and to convert local signals into a systemic defense program. In this review, we showed that the Ca^2+^ signaling is accompanied by other important rapid signals, with particular reference to ROS and electrical signaling. The interdependence of these three networked events is clear in plant-biotic interactions and have been demonstrated several times in plant–insect interactions. In order to be decoded and relayed, the Ca^2+^ message requires proteins able to bind the ion and trigger the signal transduction pathways that lead to plant responses to biotic attack. We described the role of calmodulin, calmodulin-like proteins, Ca^2+^-dependent protein kinases, annexins and calcineurin B-like proteins in plant–insect interactions. An important role in the study of Ca^2+^ signaling is the ability to visualize in real-time the generation of Ca^2+^ signals both at the single cell and whole plant level. Innovative imaging techniques have been developed based on sensitive sensors and using genetically encoded indicators that open new ways in the deciphering of the Ca^2+^ signaling in plant–insect interactions.

## Figures and Tables

**Figure 1 plants-11-02689-f001:**
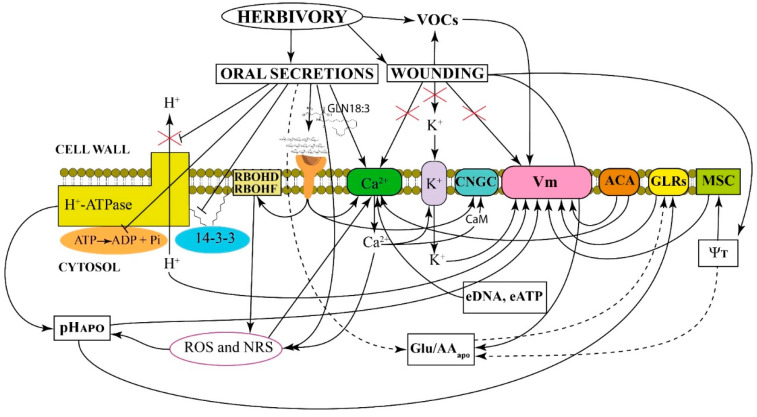
Framework of early events induced in responses to herbivory within the first minutes after infestation. Herbivory is the combination of tissue damage (wounding) and the delivery of insect oral secretions. Both herbivory and mechanical damage induce the emission of volatile organic compounds (VOCs), which depolarize the plasma membrane potential (Vm). Wounding induces variations in the water potential that affect the turgor pressure (Ψ_T_), which is perceived by plasma-membrane located mechanosensitive ion channels (MSC) resulting in Vm depolarization. Type IIB auto-inhibited Ca^2+^-ATPases (ACAs) play a role in recovery of excitability after long-term herbivory by interfering with both the Vm and [Ca^2+^]_cyt_. Mechanical wounding does not exert a direct effect on Vm, Ca^2+^ and K^+^ channel activity (red crosses). Herbivory delivers oral secretions that exert different effects. Upon herbivory receptors activate a plasma-membrane–localized cyclic nucleotide-gated ion channel (CNGC) which interacts with the Ca^2+^-sensor calmodulin (CaM). Oral secretions inhibit the H^+^-ATPase activity by hampering the association between 14-3-3 protein and H^+^-ATPase proteins and cause a reduced phosphohydrolitic activity of the proton pump with a reduced extrusion of H^+^ from the cytosol and an alkalinization of the apoplast (pH_APO_), both concurring to the Vm depolarization. Elicitors from oral secretions like *N*-linolenoyl-l-glutamine (GLN18:3) induce a RBOHD and RBOHF-dependent ROS burst. Specific oligosaccharides present in the oral secretions interact with plasma membrane receptors that also trigger ROS and Ca^2+^ signaling. Extracellular DNA (eDNA), deriving from cell disruption made by herbivory, and extracellular ATP (eATP) released by wounding and herbivory increase [Ca^2+^]_cyt_. In summary, oral secretions, wounding, Ψ_T_ reduction, pH_APO_ alkalinization, an increase in apoplastic amino acids (AAapo) and glutamate (Glu) and the activity of the glutamate receptor-like cation channels (GLRs) depolarize the Vm. Oral secretions and ROS/NRS activate the Ca^2+^ channel that in turn opens the inward rectifying K^+^ channel that reduces the cytosolic pH causing Vm depolarization. Connections supported by empirical studies in planta are represented by solid lines, hypothetical connections are shown in dashed lines.

**Figure 2 plants-11-02689-f002:**
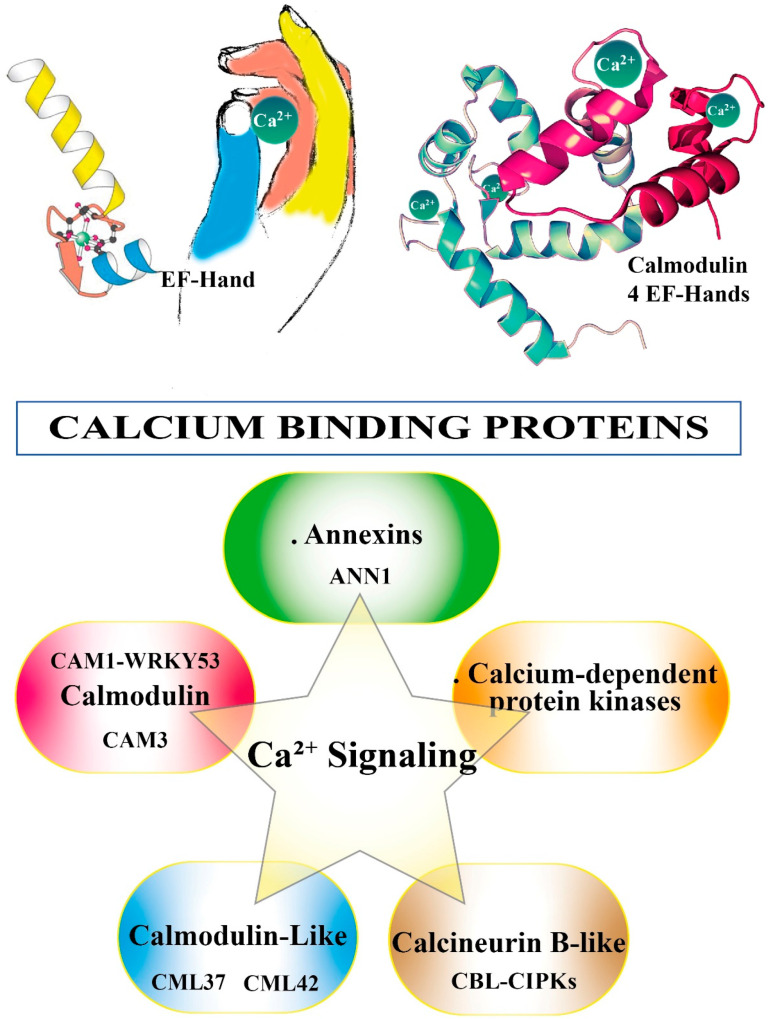
Calcium binding proteins. The top cartoon illustrates the canonical EF-hand helix–loop–helix. EF-hand Ca^2+^ binding motif contains a 29-residue helix–loop–helix topology, much like the spread thumb and forefinger of the human hand. Calmodulin is shown with its four calcium sites (shown as green balls) occupied. Calcium binding proteins are major players in Ca^2+^ signaling. See text for more details.

**Figure 3 plants-11-02689-f003:**
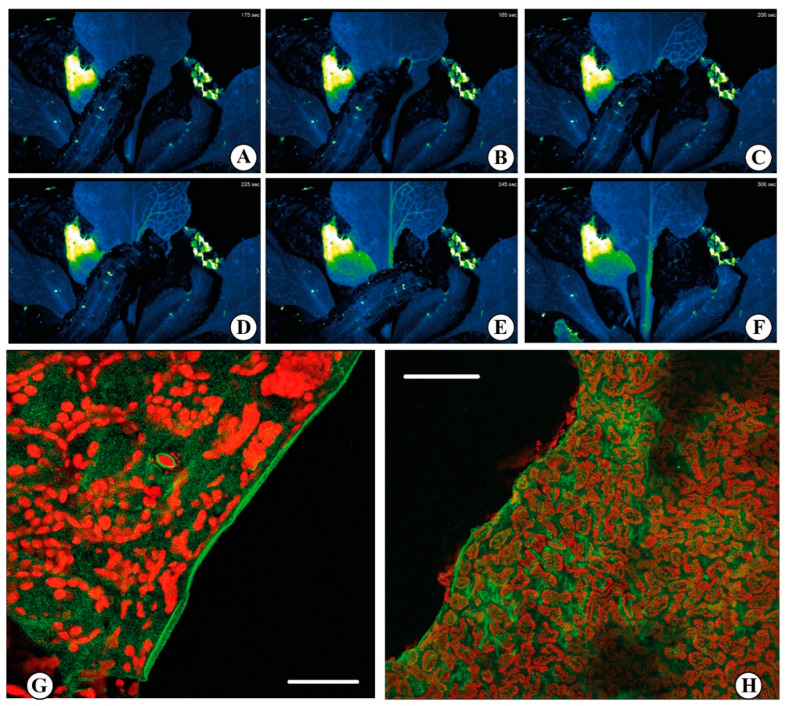
Calcium signaling upon herbivore feeding. (**A**–**F**) false colors time-course imaging of Ca^2+^ release upon feeding with the insect *Spodoptera littoralis* on a rosette leaf of *Arabidopsis thaliana* expressing the biosensor R-GECO1. The rapid fluorescence variation (blue to green) indicates the increase in the cytosolic Ca^2+^ concentration following the insect feeding. A wave of Ca^2+^ is generated at the site of feeding and rapidly spreads through the vascular system. The acquisitions have been obtained by a fluorescence stereo microscope Nikon SMZ18 with a SHR PLAN APO 0.5X WD:71. Excitation light was produced by a fluorescent CoolLED pE-300 ultra at 580 nm. Images were collected with a Mono Camera Nikon DS-Fi3 camera. Exposure time was set to 1 sec with a resolution of 8 bit 1440 × 1024. Images were acquired every 5 sec (See also Appendix A). (**G**,**H**) Confocal laser scanning micrographs showing the increased [Ca^2+^]_cyt_ in Lima bean (*Phaseolus lunatus*) leaves upon feeding by *S. littoralis*. The red fluorescence is associated to the chlorophyll present in the chloroplasts, whereas the green fluorescence indicates the cytosolic localization of Ca^2+^ by the indicator Calcium Orange^TM^. Scale bars: G = 100 µm; H = 200 µm (Figures (**G**,**H**) by Massimo Maffei).

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
