# Peer review of "Calcium Signaling in Plant-Insect Interactions"

_plants, 2022, doi:10.3390/plants11202689_

Round 1
Reviewer 1 Report
The present manuscript titled "Calcium signaling in plant-insect interactions" can be accepted for publication. The introduction provides a good understanding of the subject and its importance, with a significant quantity of information. Overall the manuscript is well designed and written, and therefore could be acceptable after revision for possible publication. Check again for some grammatical mistakes rest all look good to me.
Author Response
We thank very much the reviewer for the positive comments
The text has been carefully checked for possible grammatical errors.
Reviewer 2 Report
The review manuscript “Calcium signaling in plant-insect interactions” by Parmagnani and Maffei sent to Plants is focused on very important topic as the calcium (important secondary messenger) signaling involved in biotic stress (herbivory impact).
The manuscript is well written and cover the last advances in the topic; it will be of high interest for the scientific community working in that area.
The manuscript deserve to be published in the present form. And I would like to congrulate the authors for the well done summary.
Author Response
We are grateful to the reviewer for the comments on our manuscript.
Reviewer 3 Report
The manuscript by Parmagnani and Maffei is an updated review on calcium signalling in response to herbivory. The manuscript is well written and present many relevant aspects on this issue. I have some minor comments/concerns that should be addressed.
- The introduction section englobes a relevant aspect in calcium signalling that merits its own section, which could be named “Generation of Ca2+ signals”. In this section several information is missed and should be added. The published articles of Malabarba et al. 2021 and Meena et al., 2019 that link annexins or CNGCs with calcium signalling and herbivory should be commented. Besides, the calcium channel formed by the resistosome complex should be commented although any link with herbivory has been published yet.
- Likewise, in the Calcium and ROS signalling section, the role of RBOHD should be highlighted. In this way, the published paper of Block et al., 2018 linking oral secretions, ROS and herbivory should be commented.
- The mitogen-activated protein kinases are not calcium binding proteins. Therefore, they cannot be included in a section with this title.
Author Response
First of all we would like to thank very much Reviewer #3 for the constructive and helpful suggestions that allowed us to greatly improve our manuscript.
Below are the point-to-point answers to the reviewer's comments.
The manuscript by Parmagnani and Maffei is an updated review on calcium signalling in response to herbivory. The manuscript is well written and present many relevant aspects on this issue.
R: we thank very much the reviewer for the appreciation of our work
I have some minor comments/concerns that should be addressed.
The introduction section englobes a relevant aspect in calcium signalling that merits its own section, which could be named “Generation of Ca2+ signals”.
R: the introduction has been subdivided and now a new paragraph has a subheading "Generation of Ca2+ signals"
In this section several information is missed and should be added. The published articles of Malabarba et al. 2021 and Meena et al., 2019 that link annexins or CNGCs with calcium signalling and herbivory should be commented.
R: we thank the reviewers for these suggestions and we apologize for the overlooking of CNGCs. This new information has been added in the text and integrated in the revised Figure 1
Besides, the calcium channel formed by the resistosome complex should be commented although any link with herbivory has been published yet.
R: we thank the reviewer for this suggestion, although as a review specific with plant-insect interaction we prefer not to report data which have not been published on this topic. Despite the interesting role of the resistosome complex in Ca2+ signaling, as the reviewer pointed out no data are present upon herbivory, therefore we would prefer not add this information. On the other hand, we dedicated a new paragraph to annexins in the section related to calcium-binding protein. We apologize for overlooking this important class of CBP and we thank very much the reviewer for raising this important issue. Annexins have also been integrated in the new revises Figure 2.
Likewise, in the Calcium and ROS signalling section, the role of RBOHD should be highlighted. In this way, the published paper of Block et al., 2018 linking oral secretions, ROS and herbivory should be commented.
R: the important role of RBOHD and F have been added and discussed in the text and integrated in Figure 1 which also reports the presence of the GLN18:3 elicitor along with oligosaccharides.
The mitogen-activated protein kinases are not calcium binding proteins. Therefore, they cannot be included in a section with this title.
R: the reviewer is right, MAPK have been taken out of the context of CBP and a new paragraph now describes this class of proteins connected to Calcium signaling and herbivory.